# Awareness, Risk Perception, and Stress during the COVID-19 Pandemic in Communities of Tamil Nadu, India

**DOI:** 10.3390/ijerph17197177

**Published:** 2020-09-30

**Authors:** Jinyi Kuang, Sania Ashraf, Upasak Das, Cristina Bicchieri

**Affiliations:** 1Center for Social Norms and Behavioral Dynamics, University of Pennsylvania, Philadelphia, PA 19104, USA; saniashraf@gmail.com; 2Global Development Institute, University of Manchester, Manchester M13 9PR, UK; upasak.das@gmail.com

**Keywords:** COVID-19, coronavirus outbreak, pandemic, risk perception, awareness, India, lower-middle-income country

## Abstract

The health and economic consequences of the COVID-19 pandemic is expected to disproportionately impact residents of lower-middle income countries. Understanding the psychological impact of the pandemic is important to guide outreach interventions. In this study, we examined people’s awareness of COVID-19 symptoms, risk perception, and changes in behaviors and stress levels during the lockdown in peri-urban Tamil Nadu India. Field workers conducted phone call surveys (included *n* = 2044) in 26 communities from 20–25 May 2020. The majority perceived no (60%) or low (23%) level of risk of personally contracting coronavirus. Common fears were related to health and economic concerns, including loss of income (62%), inability to travel freely (46%), and becoming sick (46%). Residents were well aware of the common symptoms of COVID-19, such as fever (66%) and dry cough (57%), but not the asymptomatic transmission (24%). The majority experienced increased stress about finance (79%) and the lockdown (51%). Our findings emphasize the need to develop context-adequate education and communication programs to raise vigilance about asymptomatic transmission and to sustain preventative behaviors. The evidence on fear and changes in stress levels could inform designing coping strategies and programs focused on mental well-being.

## 1. Introduction

On 11 March 2020, the World Health Organization (WHO) declared the novel coronavirus outbreak (COVID-19) a global pandemic [1]. Originating from Wuhan, China, COVID-19 has spread to most countries in the world, with its impact spanning health, economics, human behaviors, and mental well-being [1,2]. As the number of deaths increases, evidence regarding people’s psychological reactions to this global public health crisis becomes increasingly important because it provides insights that could help policy-makers and practitioners improve health communication, promote preventive behaviors, and provide social and emotional support to those in need [2,3,4]. Previous studies showed that under ecological threats such as the pandemic, the negative emotional response, including increased risk perception and fear, make threats appear more menacing. The widespread fear and high risk perceptions not only leads to mental distress such as highlighted stress and anxiety, but also drive protective behavior change in response to the threats [3]. In the case of COVID-19, a rapidly growing body of literature shows that awareness of the COVID-19 symptoms, self-perceived risk of contracting diseases, and fear are salient predictors of adopting preventive behaviors [5,6,7]. For example, Harper et al. (2020) and colleagues reported that the fear of COVID-19 was the only salient predictor of positive behavior change, such as social distancing, among samples residing in the UK [6]. Wise et al. (2020) found that engagement in social distancing and handwashing was most strongly predicted by the self-perceived risk of contracting the virus among the study samples in the United States over the first week of the pandemic (11–16 March 2020) [7]. Studies examined these psychological factors and their impact on human behavior with a focus on populations residing in middle- and high-income countries [4,8,9,10,11,12,13,14]. Yet, evidence from close-knit communities where people who reside in close proximity and have strong social ties [15] in low- or lower-middle-income countries is still limited. In particular, residents from these communities can potentially be more vulnerable to the impact of the crisis due to their limited access to high-quality health-care services, high population density, large-household size, and existing financial burdens [16,17,18]. Economic disadvantage is often associated with pre-existing health conditions such as diabetes, chronic heart, and lung disease, which exposes these populations to a greater risk of death [19]. Thus, there is an increasing need to reach populations in low- or lower-middle-income countries to better understand the psychological and behavioral impact of COVID-19, as well as to inform culturally and context-appropriate policies.

In this study, we focused on communities located in the peri-urban areas of Tamil Nadu, India. According to the Indian Census 2011, peri-urban refers to areas at the periphery of cities and are formally referred to as census towns [20]. As spaces in transition, peri-urban areas in India include large numbers of migrant populations, resulting in overpopulation [21,22]. Previous studies suggest that communities in peri-urban areas have relatively dense and multiplex social networks [23]. The majority of its residents also live in poverty and are under the threat of food insecurity and unimproved sanitation conditions [24]. At the time that this survey was conducted, there have been at least 144,941 confirmed cases in India with 4171 death cases, and 17,082 confirmed cases in Tamil Nadu with 118 death cases [25]. The government of India has initiated a live update system, which includes infomercials and text messages, to disseminate COVID-19-related information and to raise awareness of the public health crisis [25]. As a preventive measure against the COVID-19 pandemic, the government of India ordered a nationwide lockdown on 24 March 2020, limiting the movement of its 1.3 billion residents [26]. This complete lockdown was implemented across the country till May 30, 2020 after which the restriction started getting partly relaxed in phases. The economic loss incurred by enforcing a lockdown in the country has threatened its growth. According to the Centre for Monitoring of Indian Economy (CMIE), India has experienced a surge in unemployment by more than 23.5% from March to April 2020 [27].

Along with the enormous economic impact, the lockdown could also impact residents’ mental health. During the time of surveys, the ongoing nationwide lockdown only allowed residents to leave their household at certain hours of the day for essential activities (e.g., seeing a health-care provider, obtaining necessary household supplies such as food). Loss of employment, restriction on non-essential activities, and social isolation could abruptly interrupt residents’ daily routine and risk their mental well-being [2]. A recent systematic review on the mental health impact of the COVID-19 outbreak suggested that the available literature was only from a limited number of countries, primarily focused on the upper-middle- and high-income countries and may not reflect the psychological response to COVID19 elsewhere around the world [28]. Therefore, insights from data collected in low- or lower-middle-income countries regarding the psychological and behavioral reactions to COVID-19 are required to address the current knowledge gap, and inform appropriate health communication, education and mental well-being programs at a local level.

In this study, we examined residents’ awareness of COVID-19 symptoms, perceived risk of contracting coronavirus, fear related to the COVID-19 pandemic, and their relationship with changes in behaviors and stress since the lockdown. Specifically, we aimed to answer the following questions: (1) what COVID-19 symptoms are people well aware of; (2) what demographic factors are related to the perceived risk of contracting coronavirus; (3) what do people fear most about the COVID-19 pandemic; (4) how risk perception and fear are related to changes in behavior and stress levels.

## 2. Materials and Methods

### 2.1. Data Collection

We conducted a cross-sectional phone survey in communities from ten census towns of peri-urban areas in Tamil Nadu India from 20–25 May 2020. During country-level lockdowns, traditional survey methods such as household visits and in-person interviews to reach residents in communities in lower-middle-income countries such as India were restricted. Online web survey techniques are often constrained by the internet coverage and literacy gap (Dillman and Bowker, 1999), which could potentially explain the current knowledge gap in the psychological response among residents in these regions. Given these constraints, rapid phone call surveys provided opportunities for data collection in countries with high phone coverage such as India, to capture the psychological and behavioral response to the COVID-19 pandemic.

The conventional samples are from an ongoing cluster-randomized trial aimed to promote improved sanitation conditions in peri-urban communities of Tamil Nadu [29]. The census towns included in this study are from two districts, Pudukkottai and Karur. The included communities are randomly selected wards from the census towns. Field workers collected phone numbers from members of these communities who had previously agreed to be contacted by the study team during the intervention period prior to the pandemic. Only one phone number per household was included to compile a sampling frame of 2657 phone numbers. The survey was translated into the local language (Tamil) by a group of bilingual researchers to ensure it fit the local context and was easy to comprehend. A total of twenty field workers from the local NGO Swasti received training remotely regarding phone survey techniques, survey instructions, knowledge about COVID-19, and standardized responses to participants’ inquiries for clarification. The total length of the phone survey took about 20 min. All participants provided oral consent at the beginning of the call. All data were anonymized for analysis. This study was reviewed and approved by the University of Pennsylvania Institutional Review Board (Protocol #: 833854).

### 2.2. Measurements

To assess participants’ awareness about COVID-19 symptoms, we asked an open-ended question: “What symptoms does one get if they have the coronavirus?” We then measured participants’ perceived risk of personally contracting coronavirus [7]. Contracting coronavirus means people have been infected with the virus, including some who are not sure how or where they became infected. The answers were categorized as no risk, low, medium, and high risk. To measure participants’ fear about coronavirus, we asked: “People in your community may fear about different aspects of how the coronavirus will affect them. What are your fears about the coronavirus?” [6] Based on qualitative discussions, we categorized respondent’s answer as self might get sick or die, family members might get sick or die, infecting other people, no cure for the disease, loss of job/income, crime increase in the community, food shortage, water shortage, police actions, inability to travel freely, social isolation/people avoiding me. Finally, we measured the change in behaviors and stress levels due to the COVID-19 lockdown. The behavioral measures included attending social gatherings, keeping a distance of six feet (social distancing), informing others about COVID-19 (i.e., talking with neighbors, friends or acquaintances about COVID-19 and sharing the information they learned related to COVID-19 such as symptoms and case updates), using cell phones/online services, and doing housework (e.g., cooking, cleaning). We reduced the stress measure to two items due to the concern of response fatigue and participants dropping out. These two items captured participants reported stress about finance, and stress about the lockdown (i.e., have to stay inside).

### 2.3. Analysis

We calculated descriptive statistics of the perceived risk of contracting coronavirus, fear of COVID-19, change in behavior, and stress level in response to the pandemic. We used Pearson’s chi-squared tests for multiple comparisons among demographic groups including gender, age, and education attainment. We used logistic regression to examine the demographic factors associated with perceiving a risk of contractive COVID-19 (coded as a binary variable, none = 0, low/medium/high = 1). We coded each preventive behavior measure (i.e., reducing social gathering, increasing social distancing and informing others about COVID-19) and stress measure (i.e., stress about finance and stress about lockdown) as binary dependent variables (0 = not practicing preventive behaviors/did not report increased stress, 1 = increased practicing preventive behaviors/report increased stress), and used logistic regression to assess each of their relationships with psychological reactions (risk perception and different types of fear) and demographic factors (i.e., gender, age groups, and education attainment). All logistic regression models were adjusted for the community-level fixed effect. All statistics were obtained at the 95% confidence interval. Data were analyzed with R 3.6 (R Foundation for Statistical Computing, Wien, Austria).

## 3. Results

Out of 2657 dialed numbers, a total of 2330 respondents provided responses to the phone survey (87.7%). To account for the community-level fixed effect, we included 2044 respondents (female = 46%) from 26 communities that had more than 30 valid responses in our analysis. The average age of respondents was 44 (SD = 14.5) years, ranging from 13 to 90 years. Among them, 15.0% have received university-level education or above, 27.7% have complete high school level education (10–12 years of schooling), 20.7% have received secondary level education (5–10 years of schooling), 17.8% have finished primary school (1–5 years of education) and 18.7% did not receive any formal education. On average, four people lived in the same household at the time of the survey (SD = 1.5, min = 1, max = 13).

### 3.1. Awareness of COVID-19 Symptoms

Of the 2044 participants, the majority were well aware of the COVID-19 symptoms, including fever (66%), dry cough (57%), difficulty breathing (50%), and other flu-like symptoms (72%). However, only 24% of respondents mentioned contracting coronavirus can have no symptoms (Table 1). Bivariate analysis shows that those who received a higher level of education (5+ years of schooling) were more likely to mention the asymptomatic features of COVID-19 than those who received no education. However, this difference did not reach statistical significance controlling for gender, age group, and community of residence (all *p* > 0.65). Males were slightly more likely than females to mention the asymptomatic feature of COVID-19 (27.4% vs. 20.2%). However, this difference also did not reach statistical significance (OR = 1.26, 95% CI: 0.96–1.66, *p* = 0.092).

### 3.2. Perceived risk of Contracting Coronavirus and Fear of Coronavirus

Out of 2044 respondents, the majority perceived that they are at a no (60.4%) or low (23.4%) risk of contracting coronavirus. Only 8.7% perceived a medium risk, and 7.6% perceived a high risk. The majority reported the fear of losing job/income (62.1%), followed by being unable to go out or travel freely (46.1%), self getting sick (45.7%), food shortage (39.2%) and family members getting sick (37.5%) (Table 2). People who received formal education (primary, secondary, high school, or university level) were more likely to perceive the risk of contracting coronavirus than those who did not receive any level of education (all *p* < 0.01) (Table 3). In general, older people were more likely to perceive the risk of contracting coronavirus compared to people younger than 30. However, these differences did not reach significance (all *p* > 0.18). We also found that, on average, males were slightly less likely than females to perceive the risk of contracting coronavirus, yet this difference did not reach statistical significance, controlling for age groups and education (OR = 0.85, 95% CI: 0.65–1.10, *p* = 0.21).

### 3.3. Change in Behavior and Stress after the Lockdown

Of the 2044 participants, the majority reported changing their daily routines in response to the pandemic, including reducing social gatherings (73.8%), increasing social distancing (80.7%), and informing others about COVID (70.2%). They also reported an increase in cell phone/internet use (61.2%) and increasing housework (e.g., cooking, cleaning) (66.9%). A large proportion of participants also reported increased stress about finances (78.7%), as well as increased stress about the lockdown (50.5%) (Table 4).

### 3.4. Relationship between Perceived Risk and Changes in Behavior and Stress Levels

We found those who perceived the risk of contracting coronavirus were more likely to inform others about COVID-19 (OR = 1.94, 95%CI: 1.46–2.59, *p* < 0.001) (Table 5). However, perceiving the risk of contracting coronavirus was only marginally correlated with social distancing (OR = 0.71, 95%CI: 0.51–0.99, *p* = 0.45) and was not significantly correlated with reducing social gatherings (OR = 0.83, 95%CI: 0.6–1.13, *p* = 0.24). We also found that those who perceive the risk of contracting the coronavirus are more likely to feel stressed about lockdown as compared to those who perceive no risk of contracting coronavirus (OR = 3.43, 95%CI: 2.58–4.57, *p* < 0.001). Those who received a high-school-level (OR = 1.86, 95%CI: 1.17–2.97, *p* = 0.01) or university-level education (OR = 1.84, 95%CI: 1.11–3.05, *p* = 0.02) are more likely to report reducing social gatherings than those who did not receive any education. We did not find sufficient evidence that engaging in preventive behavior significantly differed across gender and age groups.

We found those who perceived a risk of contracting coronavirus were more likely to report an increased stress level due to the lockdown and having to stay inside their home (OR = 3.43, 95%CI: 2.58–4.57, *p* < 0.001). Those who received no education, or a primary level of education were more likely to experience increased stress about finances and staying inside than those who have received a high school level or university level education (all *p* < 0.03). Younger people (<30 years) were also more likely to experience increased stress about finance and lockdown than people from older age groups.

## 4. Discussion

In our study, the majority of the respondents in peri-urban communities reported perceiving no or a low level of risk of personally contracting coronavirus. During the time of the survey, there was a low number of reported cases in our study regions (62 reported cases in Pudukkottai district with one death, and 95 reported cases in Karur district with no deaths) [30]. Rather than underestimating the risk as reported previously in countries such as the United States [7], our finding suggests participants might have a relatively accurate risk perception. We also found a high prevalence of fear related to both the health and economic impact of the pandemic. This is consistent with a recent study from Bangladesh, where residents reported similar mental health stressors such as fear of the outbreak, fear of personal and family members contracting coronavirus and increased stress [31].

We found that residents were well aware of the common symptoms of COVID-19, which reflected recall of critical messages included in the current COVID-19 focused information programs. Yet, only a small proportion mentioned the asymptomatic features of COVID-19, which suggests a gap in emphasizing the important role of transmission of coronavirus via asymptomatic patients. Asymptomatic patients have been proven to be contagious, which challenges transmission control, especially in communities with close social ties [32,33,34]. Such low awareness of the asymptomatic features of coronavirus might lead to inconsistently practicing preventive behaviors, such as mask wearing, social distancing, and leaving the house for non-essential activities, which can inadvertently expose vulnerable people. We also found that, despite the majority reporting positive behavior change in response to the pandemic, including reducing social gatherings, increasing social distancing, informing others of COVID-19, and other daily activities, including increasing use of phone/online services and increasing house chores, there was a considerable proportion that did not increase their preventive behaviors since the lockdown. To raise awareness of COVID-19 transmission and promote adoption of preventive behaviors, we suggest targeted education and communication programs that emphasize the role of asymptomatic transmission of coronavirus. These can be delivered through various channels (e.g., TV and radio programs) to reach households of any age or education. Specifically, the increased use of phone/online services reported in this study provides opportunities to deliver targeted messages, such as reminders to reinforce the positive preventive behaviors (e.g., habitual handwashing, reduced social interactions) and to raise vigilance about COVID-19 transmission [35,36,37].

Aside from the change in behaviors in response to the COVID-19 pandemic, we found that the majority of respondents reported experiencing increased stress about finance and about lockdown across gender, age groups, and level of education. This might potentially be due to the fact that a large proportion of the households in peri-urban areas are involved in informal labor and Micro, Small, and Medium Enterprises (MSMEs), which are particularly vulnerable to the distress caused by the lockdown [38]. Additionally, the lockdown has led to a massive loss of urban employment. Without adequate social protection, this has resulted in a painful exodus of migrants with millions of workers fleeing cities and walking hundreds of miles home [39]. These reactions could indicate limited social protection measures against economic shocks that the country offers. As reported previously, the majority of the laborers engaged in informal jobs did not have sufficient food and received minimal or no support from the government [40]. Only 6% were found to have received full wage during the period of lockdown, while 78% were found not to have received any wage [40]. Furthermore, the tremendous mental stress caused by the virtual standstill of the economy not only comes from the loss of employment, but also from the reduced chances of being employed. An expansion of social protection programs, including the public distribution system, cash transfers, and urban public works programs analogous to the existing rural employment guarantee program in India, can mitigate the risk of financial stress [41]. Furthermore, we recommend incorporating stress reduction and coping strategies in mass communication programs to address and improve residents’ mental health and well-being during the pandemic [2].

Along with the programs implemented by the government, community-based programs could also play an important role in raising awareness of COVID-19, encouraging adherence to desired norms, and protecting the mental health of community members who fall ill or have been exposed to known cases [42]. Previous studies focusing on HIV/AIDS and Ebola epidemics found that outreach through local community organizations not only improved the uptake of HIV testing and access to treatment, but also increased awareness and lowered stigma [43,44,45]. Guided by these relevant experiences, we suggest community-led participation programs through non-contact channels such as phone or web service to collectively respond to the COVID-19 pandemic. These programs could include mutually checking on neighbors’ wellbeing, using social media to reach network members and community support, and coping programs to ensure sustainable and inclusive participation. We also encourage more community-based studies to better understand the range of psychological and behavioral reactions to COVID-19 pandemic, including the difficulties caused by government policies and mental health risks that might have been underestimated.

This study has several limitations. First, this exploratory study does not represent the psychological and behavioral reactions at a state or national level. Yet, the evidence regarding residents’ psychological and behavioral reactions to COVID-19 at the specific time point could be used to guide program design in similar study contexts. Second, the phone call survey technique has several constraints. For example, we used open-ended questions to measure respondents’ awareness of COVID-19 symptoms. It is possible that respondents were aware of the asymptomatic feature but forgot to recall it during the call. Other caveats related to using phone calls include lack of representativeness and length of time, which can limit the level of engagement from the respondent [46,47,48,49]. However, given the local context of the study location and COVID-19-related policies in place in these regions, a rapid phone call survey was an adequate method to understand the current conditions.

## 5. Conclusions

To conclude, this study provides evidence regarding awareness of COVID-19 symptoms, risk perception, fear, stress, and changes in behavior in response to the COVID-19 pandemic from close-knit communities in peri-urban India. We highlighted potential knowledge gaps (e.g., low level of awareness of the asymptomatic features) among residents, as well as the common fears and sources of stress since the lockdown. Our findings may shed light on developing context-adequate education and communication programs to raise awareness of coronavirus transmission routes and promote preventive behaviors. The evidence about fears and stress may serve as a reference in designing coping strategies and programs with a focus on mental well-being.

## Figures and Tables

**Table 1 ijerph-17-07177-t001:** Awareness of COVID-19 symptoms by gender and education, Tamil Nadu, India, May 2020.

Knowledge of COVID-19 Symptoms n(%)	Total (*n* = 2044)	Female (*n* = 954)	Male (*n* = 1090)	No School (*n* = 363)	Primary (1–5 Years) (*n* = 424)	Secondary (5–10 Years) (*n* = 567)	High School (10–12 Years) (*n* = 383)	University (12+) (*n* = 307)
Can have no symptoms	492 (24.1%)	193 (20.2%)	299 (27.4%)	56 (15.4%)	65 (15.3%)	153 (27.0%)	119 (31.1%)	99 (32.2%)
Fever	1356 (66.3%)	583 (61.1%)	773 (70.9%)	191 (52.6%)	259 (61.1%)	393 (69.3%)	273 (71.3%)	240 (78.2%)
Dry cough	1163 (56.9%)	505 (52.9%)	658 (60.4%)	158 (43.5%)	210 (49.5%)	347 (61.2%)	241 (62.9%)	207 (67.4%)
Difficulty breathing	1031 (50.4%)	465 (48.7%)	566 (51.9%)	151 (41.6%)	175 (41.3%)	303 (53.4%)	207 (54.0%)	195 (63.5%)
Decreased smell/taste	268 (13.1%)	121 (12.7%)	147 (13.5%)	23 (6.3%)	33 (7.8%)	80 (14.1%)	64 (16.7%)	68 (22.1%)
Other flu like symptom	1470 (71.9%)	699 (73.3%)	771 (70.7%)	238 (65.6%)	274 (64.6%)	429 (75.7%)	300 (78.3%)	229 (74.6%)
Headaches	385 (18.8%)	157 (16.5%)	228 (20.9%)	39 (10.7%)	90 (21.2%)	103 (18.2%)	73 (19.1%)	80 (26.1%)

Note: In addition to the commonly mentioned symptoms, 73 (3.6%) mentioned blindness, 35 (1.7%) mentioned stroke, and 82 (4.0%) mentioned heart attack.

**Table 2 ijerph-17-07177-t002:** Perceived risk of contracting coronavirus and fear of COVID-19 by gender and age group across study communities, Tamil Nadu, India, May 2020.

Factors	Total (*n* = 2044)	Female (*n* = 954)	Male (*n* = 1090)	<30 Years (*n* = 327)	30–39 Years (*n* = 456)	40–49 Years (*n* = 535)	50–59 Years (*n* = 363)	60 Years or Above (*n* = 363)
Perceived risk of contracting coronavirus *n* (%)								
High	155 (7.6%)	49 (5.1%)	106 (9.7%)	8 (2.4%)	19 (4.2%)	62 (11.6%)	44 (12.1%)	22 (6.1%)
Medium	177 (8.7%)	88 (9.2%)	89 (8.2%)	46 (14.1%)	35 (7.7%)	43 (8.0%)	25 (6.9%)	28 (7.7%)
Low	478 (23.4%)	220 (23.1%)	258 (23.7%)	60 (18.3%)	101 (22.1%)	111 (20.7%)	93 (25.6%)	113 (31.1%)
No risk	1234 (60.4%)	597 (62.6%)	637 (58.4%)	213 (65.1%)	301 (66.0%)	319 (59.6%)	201 (55.4%)	200 (55.1%)
Fears related to COVID-19 *n* (%)								
Loss of job/income	1269 (62.1%)	611 (64.0%)	658 (60.4%)	223 (68.2%)	296 (64.9%)	334 (62.4%)	205 (56.5%)	211 (58.1%)
Inability to travel freely	942 (46.1%)	430 (45.1%)	512 (47.0%)	155 (47.4%)	190 (41.7%)	257 (48.0%)	169 (46.6%)	171 (47.1%)
Self might get sick	934 (45.7%)	406 (42.6%)	528 (48.4%)	140 (42.8%)	199 (43.6%)	254 (47.5%)	176 (48.5%)	165 (45.5%)
Food shortage	802 (39.2%)	380 (39.8%)	422 (38.7%)	127 (38.8%)	197 (43.2%)	215 (40.2%)	134 (36.9%)	129 (35.5%)
Family members might get sick or die	766 (37.5%)	342 (35.8%)	424 (38.9%)	134 (41.0%)	176 (38.6%)	183 (34.2%)	131 (36.1%)	142 (39.1%)
Infecting other people	572 (28.0%)	241 (25.3%)	331 (30.4%)	111 (33.9%)	134 (29.4%)	125 (23.4%)	101 (27.8%)	101 (27.8%)
There is no cure	438 (21.4%)	192 (20.1%)	246 (22.6%)	98 (30.0%)	99 (21.7%)	89 (16.6%)	80 (22.0%)	72 (19.8%)
Dying	301 (14.7%)	114 (11.9%)	187 (17.2%)	50 (15.3%)	66 (14.5%)	68 (12.7%)	56 (15.4%)	61 (16.8%)
Water shortage	253 (12.4%)	91 (9.5%)	162 (14.9%)	53 (16.2%)	59 (12.9%)	80 (15.0%)	34 (9.4%)	27 (7.4%)
Police actions	227 (11.1%)	86 (9.0%)	141 (12.9%)	56 (17.1%)	51 (11.2%)	52 (9.7%)	31 (8.5%)	37 (10.2%)
Increased crime in the community	202 (9.9%)	74 (7.8%)	128 (11.7%)	43 (13.1%)	54 (11.8%)	45 (8.4%)	28 (7.7%)	32 (8.8%)
Social isolation/people avoiding me	201 (9.8%)	65 (6.8%)	136 (12.5%)	28 (8.6%)	36 (7.9%)	53 (9.9%)	39 (10.7%)	45 (12.4%)

**Table 3 ijerph-17-07177-t003:** Factors associated with the perceived risk of contracting COVID-19 adjusted for community-level fixed effects, Tamil Nadu, India, May 2020.

Independent Variables	Dependent Variable: Perceiving A Risk of Contracting Coronavirus
	Adjusted OR (95% CI)
Gender (ref. female)	
Male	0.85 (0.65–1.10)
Age group (ref. < 30 years)	
30–39 years	0.9 (0.59–1.38)
40–49 years	1.1 (0.71–1.71)
50–59 years	1.13 (0.7–1.82)
60 years or above	1.41 (0.86–2.32)
Education attainment (ref. no education)	
Primary (1–5 years)	2.16 *** (1.47–3.17)
Secondary (5–10 years)	1.96 ** (1.32–2.93)
High school (10–12 years)	2.09 ** (1.35–3.25)
University (12+)	2.57 *** (1.59–4.19)

Note: OR = Odds Ratios; ** *p* < 0.01, *** *p* < 0.001.

**Table 4 ijerph-17-07177-t004:** Change in behaviors and stress levels after lockdown by gender and age groups across study communities, Tamil Nadu, India, May 2020.

Change in Behaviors and Stress Levels *n* (%)	Total (*n* = 2044)	Female (*n* = 954)	Male (*n* = 1090)	<30 Years (*n* = 327)	30–39 Years (*n* = 456)	40–49 Years (*n* = 535)	50–59 Years (*n* = 363)	60 Years or Above (*n* = 363)
Attending social gatherings-reduced	1508 (73.8%)	672 (70.4%)	836 (76.7%)	242 (74.0%)	324 (71.1%)	398 (74.4%)	283 (78.0%)	261 (71.9%)
Keeping a distance of 2 m/6 ft-increased	1650 (80.7%)	747 (78.3%)	903 (82.8%)	254 (77.7%)	373 (81.8%)	431 (80.6%)	304 (83.7%)	288 (79.3%)
Informing others about COVID-19-increased	1435 (70.2%)	649 (68.0%)	786 (72.1%)	215 (65.7%)	334 (73.2%)	393 (73.5%)	262 (72.2%)	231 (63.6%)
Use of cell phones/online services-increased	1250 (61.2%)	541 (56.7%)	709 (65.0%)	219 (67.0%)	297 (65.1%)	334 (62.4%)	213 (58.7%)	187 (51.5%)
Doing housework-increased	1368 (66.9%)	674 (70.6%)	694 (63.7%)	213 (65.1%)	318 (69.7%)	372 (69.5%)	241 (66.4%)	224 (61.7%)
Stress about finance-increased	1608 (78.7%)	754 (79.0%)	854 (78.3%)	253 (77.4%)	372 (81.6%)	420 (78.5%)	289 (79.6%)	274 (75.5%)
Stress about lockdown/stay inside-increased	1032 (50.5%)	440 (46.1%)	592 (54.3%)	147 (45.0%)	233 (51.1%)	286 (53.5%)	186 (51.2%)	180 (49.6%)

**Table 5 ijerph-17-07177-t005:** Factor associated with changes in behavior and stress since lockdown adjusted for community-level fixed effects, Tamil Nadu, India, May 2020.

Factors	Reduce Group Gathering	Increase Social Distancing/Stay 6 Feet Away from Others	Increase Informing Others about COVID-19	Increase Stress About Staying Inside	Increase Stress About Finances
	Adjusted OR (95%CI)				
Perceived risk of contracting coronavirus (ref. no risk)					
Any level of risk	0.83 (0.6–1.13)	0.71 * (0.51–0.99)	1.94 *** (1.46–2.59)	3.43 *** (2.58–4.57)	0.96 (0.7–1.3)
Gender (ref. female)					
Male	1.1 (0.84–1.44)	1.32 (0.99–1.75)	1.09 (0.85–1.4)	1.15 (0.9–1.47)	0.99 (0.75–1.31)
Age group (ref. < 30 years)					
30–39 years	0.88 (0.58–1.34)	0.98 (0.64–1.51)	1.21 (0.81–1.79)	1.01 (0.68–1.5)	0.85 (0.55–1.32)
40–49 years	1.13 (0.72–1.76)	0.76 (0.49–1.18)	1.16 (0.77–1.76)	0.86 (0.57–1.3)	0.59 * (0.37–0.92)
50–59 years	1.17 (0.7–1.94)	1.06 (0.64–1.77)	1.02 (0.64–1.62)	0.64 (0.41–1)	0.57 * (0.34–0.94)
60 years or above	1.05 (0.63–1.76)	0.8 (0.47–1.35)	0.59 * (0.36–0.95)	0.5 ** (0.31–0.8)	0.39 *** (0.23–0.65)
Education attainment (ref. no education)					
Primary (1–5 years)	1.28 (0.83–1.96)	1 (0.63–1.58)	1 (0.69–1.46)	1.47 * (1.02–2.13)	2.28 *** (1.48–3.52)
Secondary (5–10 years)	1 (0.66–1.51)	1.36 (0.88–2.12)	1.09 (0.75–1.58)	0.91 (0.63–1.31)	1.19 (0.78–1.8)
High school (10–12 years)	1.86 ** (1.17–2.97)	1.39 (0.85–2.26)	1 (0.66–1.52)	0.57 ** (0.37–0.86)	0.57 * (0.36–0.89)
University (12 +)	1.84 * (1.11–3.05)	1.45 (0.86–2.46)	1.64 * (1.02–2.65)	0.56 * (0.36–0.89)	0.56 * (0.33–0.93)

Note: OR = Odd Ratios; * *p* < 0.05; ** *p* < 0.01, *** *p* < 0.001.

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
