# Peer review of "Awareness, Risk Perception, and Stress during the COVID-19 Pandemic in Communities of Tamil Nadu, India"

_ijerph, 2020, doi:10.3390/ijerph17197177_

Round 1
Reviewer 1 Report
This research conducted a phone call survey to measure the awareness, perceived risk and stress of COVID-19 pandemic in suburban area of India.
However, the result are mainly the accumulation of descriptive data analysis, and the statistical inference does not provide novel insights.
This research need a deep analysis of the peri-urban India, otherwise the result could not be generalized to lower-middle income countries and could not shed any light beyond Tamil Nadu, India.
Line 166, should be "food shortage".
Reviewer 2 Report
This manuscript presents a relevant and valuable topic and referred to a study area that has never been studied. The research was fulfilled with solid empirical investigation, as well as detailed analysis. I recommend publications with some minor details.
Firstly, in the Introduction, it could be helpful to clarify when the lockdown ends. In this way, Section 3.3. would be clearer to the reader.
Secondly, the authors could indicate the software used for the descriptive statistical analysis. It would allow the Analysis section to be more complete (lines 126-127).
Finally, Section 3.1 would be more complete with a brief description of what emerged among participants with different education levels. From Table 1 some differences emerge in recognizing the coronavirus symptoms, try to emphasize them.
Reviewer 3 Report
Review Report
Dear Authors
The harm of the COVID-19 pandemic is a serious and serious issue. I am very pleased to have more scholars and experts discussing public health issues. The results of this manuscript will help to understand intensive community public health prevention and infection measures. However, the publication of research results requires more reinforcement. The following suggestions are provided, and the author is requested to reply:
- In the Introduction narrative, the author emphasized the seriousness of COVID-19 and the dense population of the country under study. This is a very good start. However, the title stated that the authors would like to use Risk Perception, and Stress During as the subject to investigate the feelings of the research subjects. In the Introduction article, the research results are mentioned, but the details of the research investigation cannot be explained. This importance will affect the establishment of research tools in the Materials and Methods chapter. Please improve and add a description.
- Usually at the beginning of the manuscript, there will be clear information explaining the historical research results or the source of the research topic, and it is recommended to add explanations. This will make the manuscript more complete.
- Good research must not only have sufficient explanation, but also a clear research context and reliable analysis objects. It must also clearly explain what theories and research methods are used in the research. However, as stated in the summary in the manuscript (line 16), although the research team stated, the study sampled 2044 samples. However, in 2. Materials and Methods, the manuscript only provides the sampling process and the expected total number of samples, and lacks the result of the final sampling number.
- Good research results need to be presented in a scientific way. There are many ways to present it. However, in the manuscript, it seems impossible to clearly explain the main structure of the research, what research theory and research methods are used for data analysis. This will be a major test for the presentation of future scientific research results and reference value of the manuscript. (line 87-138 ?)
- Scientific investigations require rigorous basis, and the compilation of questionnaire tools must also be based on detailed references. The source of the questionnaire used in the manuscript needs to be explained and explained in order to obtain more reliable information for readers to read and provide future scientific research references (in Chapter Materials and Methods).
On the whole, this article is a very meaningful research topic. I hope that the authors will improve the COVID-19 research after improving the problem.
wish all the best,

Reviewer 4 Report
Dear Authors,
I would like to congrats wit you for this valuable article.
I believe that it significantly contributes to increasing our understanding of the perception of COVID-19.
I have some minor comments which I hope can improve your article:
1. Could you provide more details about the RCT study?
2. Could you specify if the questionnaire is anonymized?
3. Could you give more information about the behavioural measure "information others about COVID-19" (line 120-21)?
4. I do not understand why you have inserted "blindness, stroke and heart attack" in table 1 (line 160)
5. what do you mean with "contracting coronavirus"? do you included in this definition also asymptomatics?
Thank you for your effort to answer my questions.
Reviewer 5 Report
This purpose of this study was to explore the level of awareness, risk perception, fear and behavioral changes in the peri-urban areas of Tamil Nadu, India during the COVID-19 outbreak. The topic is interesting and presents a good degree of novelty. Overall, the manuscript is well written, the results are clearly described and adequately discussed, while the conclusions support the main findings reported.
Minor comments.
Line 84. Consider adding “what” before “are people well-aware…”.
Lines 154-159. I could not find this data in any table.
Round 2
Reviewer 1 Report
Thank you for your hard working. I could see the improvement of this paper. The reply also make sense to me. Therefore, I would recommend to publish.
Reviewer 3 Report
Dear author
This is a meaningful research report, and I believe it can be helpful to the actual situation and the current research situation.
Therefore, I recommend that this research report be accepted.